# Genetic Diversity and Population Structure of the Invasive Oriental Fruit Fly, *Bactrocera dorsalis* (Diptera: Tephritidae) in Burkina Faso

**DOI:** 10.3390/insects15050298

**Published:** 2024-04-23

**Authors:** Nouhoun Traoré, Mahamadi Kientega, Hamidou Maïga, Karim Nebié, Issaka Zida, Roberto Galizi, Emmanuel Kiendrebeogo, Bazoumana B. D. Sow, Adrien M. G. Belem, Remy A. Dabiré, Abdoulaye Diabaté

**Affiliations:** 1Institut de Recherche en Sciences de la Santé (IRSS), Bobo-Dioulasso 01 BP 545, Burkina Faso; mkient54@gmail.com (M.K.); maigahamid@yahoo.fr (H.M.); emmanuelkiendrebeogo0@gmail.com (E.K.); sowbazoumana@gmail.com (B.B.D.S.); npiediab@gmail.com (A.D.); 2Laboratoire de Santé Animale Tropicale, Institut du Développement Rural, Université Nazi Boni (UNB), Bobo-Dioulasso 01 BP 1091, Burkina Faso; belemamg@hotmail.fr; 3Institut de l’Environnement et de Recherches Agricoles (INERA), Bobo-Dioulasso 01 BP 910, Burkina Faso; nebkar87@gmail.com (K.N.); ishakazida@gmail.com (I.Z.); dabire_remy@yahoo.fr (R.A.D.); 4Centre for Applied Entomology and Parasitology, School of Life Sciences, Keele University, Staffordshire ST5 5BG, UK; r.galizi@keele.ac.uk

**Keywords:** genetic diversity, genetic structure, microsatellite markers, *Bactrocera dorsalis*, Burkina Faso

## Abstract

**Simple Summary:**

*Bactrocera dorsalis* is a highly invasive fruit fly that is of major economic importance worldwide. In Burkina Faso, it is one of the main problems associated with mango production, and it is often responsible for the loss of the whole harvest in the most affected areas. Innovative strategies are being developed to improve the current means of controlling this pest. The aim of this study was to understand the distribution, diversity, and genetic structure of *B. dorsalis* in Burkina Faso. Male *B. dorsalis* were collected transversally in Burkina Faso in July 2021. The results showed that the abundance of *B. dorsalis* varied from 87 to 2986 flies per trap per day at the different sampling sites. The genetic variability was high at all sites, with an average Shannon diversity index of 0.72 per site. The migration rates were high between the study populations and ranged from 10.62 to 27.53 migrants. A genetic structural analysis identified three weakly separated genetic groups in the population of *B. dorsalis* in Burkina Faso. The results of this study will help to better guide control interventions against this pest.

**Abstract:**

*Bactrocera dorsalis* Hendel is a highly invasive horticultural pest that is of major economic importance worldwide. In Burkina Faso, it is one of the main insect pests that affects the production and exportation of mangos. Understanding the biology and the genetic dynamics of this insect pest provides crucial information for the development of effective control measures. The aim of this study was to understand the distribution, diversity, and genetic structure of *B. dorsalis* in Burkina Faso. Male flies were collected transversally in Burkina Faso and analyzed by PCR using 10 microsatellite markers. The results showed an abundance of *B. dorsalis* varying from 87 to 2986 flies per trap per day at the different sampling sites. The genetic diversity was high at all sites, with an average Shannon’s Information Index (I) of 0.72 per site. The gene flow was high between study populations and ranged from 10.62 to 27.53 migrants. Bayesian admixture analysis showed no evidence of structure, while Discriminant Analysis of Principal Components identified three weakly separated clusters in the population of *B. dorsalis* in Burkina Faso. The results of this study could be used to optimize the effectiveness of current control interventions and to guide the implementation of new, innovative, and sustainable strategies.

## 1. Introduction

The Tephritidae family includes fruit flies, which are of major economic importance worldwide [1,2,3]. They are among the main insect pests in horticulture due to their life cycle being dependent on host plants, which provide breeding grounds and sources of nutrients for the development of the larvae [2,4,5]. They are responsible for huge economic losses resulting from crop losses in the affected countries [1,6]. The most damaging fruit fly species belong to the following genera: *Anastrepha*, *Bactrocera*, *Ceratitis*, *Dacus*, *Rhagoletis*, and *Zeugodacus* [3]. The genus *Bactrocera* comprises around 460 species organized into subgroups, including the *Bactrocera dorsalis* complex [7,8].

The *Bactrocera dorsalis* complex *sensu lato* (*s.l.*) comprises around one hundred morphologically similar species [9,10]. Most are less economically important, with the exception of *B. dorsalis* (Hendel), commonly known as the oriental fruit fly. *Bactrocera dorsalis* (Hendel) and its synonymous species, such as *Bactrocera papayae* (Drew & Hancock), *Bactrocera philippinensis* (Drew & Hancock) and *Bactrocera invadens* (Drew, Tsuruta, & White), are notorious insect pests; highly polyphagous, with over 250 host plant species; and extremely invasive. They are particularly fecund and highly resilient to environmental stresses [3,5,9,11].

*Bactrocera dorsalis* has a worldwide distribution. Its presence in Africa was detected in Kenya in 2003 and it was first described as *B. invadens* [12], a species closely related to *B. dorsalis*, as were *B. philippinensis*, described in the Philippines, and *B. papayae*, described in Thailand, Malaysia, and Indonesia [13,14,15]. *Bactrocera dorsalis* has now spread throughout sub-Saharan Africa, where its capacity for invasion and destruction has made it the main insect pest faced by the horticultural sector; it is estimated to cost the industry nearly USD 2 billion in damages per year [16,17,18,19].

In Burkna Faso, *B. dorsalis* was detected for the first time in 2005 [1,16,20]. It is now the main problem in the fruit sector, affecting a dozen cultivated and/or wild host plants, the main ones being mango (*Mangifera indica*) and shea (*Vitellaria paradoxa*). The damage to mangoes is much more significant, with crop losses of up to 100% in some orchards, and quarantine measures affect the export of fruit from infested countries, contributing to significant economic losses [1,20].

The control of *B. dorsalis* is essentially based on the use of chemical pesticides, which unfortunately pollute the environment, affecting the food chain and human well-being. Furthermore, nowadays, these pesticides are inefficient due to resistance problems developed by insect pests [17,19]. Eco-friendly techniques involving sterile insect techniques (SIT), entomopathogenic fungi, parasitoids, and predatory ants have been adopted in control programs [17,21,22]. Unfortunately, these resources are limited by their cost, accessibility, and/or the complexity of their implementation.

Recent advances in genetic engineering have led to the emergence of innovative tools for controlling insect pests. Genetic control appears to be a promising technology, which is based on the introduction of genetic traits which interfere with insects’ ability to develop or reproduce, thereby suppressing their populations [23]. These new technologies are potentially more species-specific, accessible, non-polluting, easy to deploy, and applicable on a larger scale compared to existing methods [24,25]. However, effective control of insect pests, especially through genetic control strategies, would require an understanding of genetic diversity, genetic structure, and gene flow within target populations [24,26,27].

Microsatellites are genetic markers of nuclear origin that are non-coding, codominant, and highly polymorphic. They are composed of short nucleotide sequences repeated in tandem, distributed throughout the eukaryotic genome, and are powerful tools for the genetic analysis of natural populations. Studies have characterized several microsatellite markers in *B. dorsalis* [21,28]. In this study, we exploited the potential of microsatellite markers to elucidate the genetic diversity, gene flow, and genetic structure of *B. dorsalis* populations in Burkina Faso.

## 2. Materials and Methods

### 2.1. Sample Collection

Adult *B. dorsalis* flies were collected from sites in the western part of Burkina Faso in July 2021, in three types of plant formations, including mango orchards (MO), shea agroforestry parks (SAP), and natural formations (WLD). Research found that July is the outbreak period for *B. dorsalis* in the west part of Burkina Faso. This is thought to be mainly influenced by the availability of host fruits and the suitability of climatic conditions with a favorable atmospheric humidity and temperature [1,29]. Sampling was carried out at seven different sites, including Comoé (CMOE) in the Cascades region, Boulkiemdé–Sanguié (BK-SG), Sissili (SSLI) in the Centre-Ouest region, Kénédougou (KNDG), Houet (HOUET) in the Hauts-Bassins region, Bougouriba (BGRB), and Poni (PONI) in the Sud-Ouest region (Figure 1). Two Tephri traps, implemented with a male sex-specific attractant for *Bactrocera* species, one of the most effective attractants for *B. dorsalis*, the Methyl-eugenol (ME), and an insecticide, the Dichlorvos or 2,2-dichlorovinyl dimethyl phosphate (DDVP), were placed in each plant formation at intervals of 100 m radius, for 24 h of exposure. All the flies collected were identified morphologically under the microscope using the available taxonomic identification keys [13,30], then counted and conserved in 80% ethanol for molecular analysis.

### 2.2. Molecular Analyses

DNA was extracted from 315 male flies, consisting of 15 individuals randomly sampled from each type of plant formation per site, using DNAzol (invitrogen, Thermo Fisher Scientific, Waltham, MA, USA) according to the manufacturer’s instruction. The final product was collected in 100 mL EDTA and stored at −80 °C for further analyses. Ten microsatellite markers, Bd15, Bd19, Bd85b, Bi1, Bi5, Bi8, Bi10, MS3, MS4, and MS12A, were analyzed in this study due to their polymorphism, which was reported in previous studies [5,31,32]. PCRs were performed in a total volume of 25 µL using a modified version of the protocol described by Yi et al. (2016) [33]. Briefly, the PCR mix consisted of 6 µL of the FIREPol Master Mix, 1.5 µL of each primer, 14 µL of H_2_O, and 2 µL of the DNA template. Amplifications were performed with the Thermo Fisher 5020 Arktik PCR Thermal Cycler using the following program: initial denaturation at 94 °C for 3 min, followed by 35 cycles of 94 °C for 30 s, 56–61 °C for 30 s, and 72 °C for 45 s, a final extension at 72 °C for 5 min, and final storage at 4 °C. PCR products were migrated onto a 3% agarose gel and bands were analyzed using the Benchtop UV Transilluminator 2UV.

### 2.3. Data analysis

#### 2.3.1. Flies Abundance, Genetic Diversity and Genetic Relations Analysis

Different types of software were used to analyze the data presented in this work. The R version 4.3.0 software was used in the R-studio environment to analyze the *B. dorsalis* abundance in Burkina Faso by calculating the density of flies per trap per day (FTD). GenAlEx version 6.51b2 [34] was used to summarize the statistical genetics by calculating the number of different alleles (Na), effective number of alleles (Ne), Shannon’s Information Index (I), observed heterozygosity (Ho), expected heterozygosity (He), total expected heterozygosity (Ht), unbiased expected heterozygosity (uHe), and fixation index (F) for each locus in the whole population, and for each sub-population. The Shannon Information Index or Shannon Diversity Index is used to measure the diversity at different levels, including genes, populations, whole species, and ecosystems. It is widely employed in population genetics studies as a suitable estimator with which to evaluate the variation at multiallelic loci, such as microsatellites [35]. GenAlEx version 6.51b2 was then used to determine Wright’s fixation indexes (Fis, Fit, and Fst), gene flow as number of migrants (Nm) according to each locus in the total population. It was also used to estimate genetic differences between sampling populations by analyzing the molecular variance (AMOVA), the genetic differentiation index (Fst), and the gene flow between population pairs as the number of migrants (Nm). The Nei’s genetic distance [36] was estimated between pairs of populations.

#### 2.3.2. Analysis of Genetic Structure 

The genetic structure of *B. dorsalis* populations in Burkina Faso was analyzed using inferential methods such as Bayesian admixture and discriminant analysis of principal components (DAPC) [37].

STRUCTURE software version 2.3.4 was used for the Bayesian admixture analysis. Ancestral genotypic proportions were inferred by considering a number of clusters (K) varying from 1 to 10 with 10 independent replications for each value of K, setting the burn-in period to 50,000 iterations, and considering 100,000 Markov Chain Monte Carlo (MCMC) for each value of K. The expected value of K was determined according to the method of Evanno et al. (2005) using the Structure Harvester platform developed by Earl et al. (2012) [38,39]. The graphical representation of the analysis was generated using STRUCTURE Bar plot [40].

Discriminant Analysis of Principal Components (DAPC) was used to infer genetic relationships between individuals using the *adegenet* package version 2.1.10 in R version 4.3.0 software. The reproducibility of the analysis results was ensured by setting the random number value to 123 using the *set.seed()* function. The cross-validation approach through the *xvalDapc()* function and numerical parameters such as the number of principal components (PCs) achieving highest mean success and the number of PCs achieving the lowest root mean squared error (RMSE) were used to determine the optimal number of PCs to retain in the analysis. Missing values in the data were replaced by the mean of the non-missing values. The number of clusters in the *B. dorsalis* population in Burkina Faso was determined based on the *find.clusters()* function using the Bayesian Information Criterion (BIC). The number of discriminant functions was determined based on the graph of discriminant analysis eigenvalues. The final population structure graph for *B. dorsalis* in Burkina Faso was generated from the number of PCs and the number of associated discriminant functions by scatter plotting axis 1 against axis 2.

## 3. Results

### 3.1. Abundance and Diversity

#### 3.1.1. Abundance of *Bactrocera dorsalis* Pests in Burkina Faso

The transversal sampling of *B. dorsalis* in July 2021, which took place across seven sites in Burkina Faso, revealed a differential abundance of flies caught depending on the sampling sites and the type of plant formation, including mango orchards, shea agroforestry parks, and natural formations. The highest density per site, with 2986 FTD (56.88%), was observed in Kénédougou (KNDG) while the lowest, with 87 FTD (1.66%), was observed in Sissili (SSLI). The highest density of flies per type of plant formation, with 2522 FTD (48.03%), was observed in mango orchards, while the lowest, with 1118 FTD (21.29%), was observed in natural formations. Regarding the number of flies collected per type of plant formation per site, the highest densities, with 68 FTD (43.59%) and 63 FTD (70.79%) were observed in the natural formations in Bougouriba and Boulkiemdé–Sanguié, respectively. In Comoé, Houet, Poni and Sissili, the highest densities, with 504 FTD (79.18%), 149 FTD (83.71%), 645 FTD (57.69%) and 45 FTD (51.72%) were observed in the mango orchards. In Kénédougou, the highest density 1313 FTD (43.97%) was observed in the shea agroforestry park. The lowest density of flies per type of plant formation per site was observed in the mango orchard in Bougouriba, with 38 FTD (24.36%); in the shea agroforestry park in Boulkiemdé–Sanguié and Poni, with 9 FTD (10.11%) and 92 FTD (8.23%), respectively; and in the natural formation in Comoé, Houet, Kénédougou, and Sissili, with 41 FTD (6.43%), 5 FTD (2.82%), 549 FTD (18.39%) and 11 FTD, (12.64%), respectively (Table 1).

#### 3.1.2. Polymorphism of Microsatellite Markers and Genetic Diversity

Genetic polymorphism analysis of the ten microsatellite loci in 315 individuals of the *B. dorsalis* species from seven localities revealed a total diversity of 44 alleles in the whole population. The total number of alleles per locus ranged from 3 (Bd15) to 7 (Bd85b), with an average of 4.4 per locus. The number of alleles per sampling site ranged from 28 (Sissili) to 35 (Kénédougou), with an average of 31.14 per locality (Appendix A). As shown in Table 2, the mean numbers of alleles (Na) at locus level ranged from 1.286 (Bd15) to 4.429 (MS4), with a total mean of 3.114 per locus, while the mean effective numbers of alleles (Ne) ranged from 1.007 (Bd15) to 3.131 (Bd19) with a total mean of 1.924 per locus. Shannon’s Information Index (I) ranged from 0.018 (Bd15) to 1.239 (Bd19) with an average of 0.722 per locus. The mean observed heterozygosity (Ho) per locus ranged from 0.003 (Bd15) to 0.791 (MS4), with a mean of 0.356, while the mean expected heterozygosity (He) ranged from 0.006 (Bd15) to 0.673 (Bd19), with a mean of 0.402 per locus (Appendix A). Total Expected Heterozygosity (Ht) ranged from 0.006 (Bd15) to 0.7 (Bd19) in the total population, with an average of 0.413 per locus. The hybridization coefficients Fis (Inbreeding Coefficient) and Fit (Total Inbreeding Coefficient) varied, respectively, from −0.253 (MS12A) to 0.704 (Bi8) and from −0.233 (MS12A) to 0.719 (Bi8), with averages of 0.208 and 0.228 per locus. Genetic differentiation indices (Fst) at the locus level ranged from 0.014 (Bi1) to 0.052 (Bi8), with an average of 0.027 per locus, leading to the gene flow (Nm) ranging from 4.561 (Bi8) to 17.553 (Bi1), with an average of 11.36 per locus (Table 2).

The average number of alleles (Na) per locus for each locality ranged from 2.8 (SSLI) to 3.5 (KNDG), with an average of 3.1 per locality, while the average effective number of alleles (Ne) ranged from 1.819 (SSLI) to 2.032 (KNDG), with an average of 1.924 per locality. Shannon’s Information Index (I) per locality ranged from 0.657 (SSLI) to 0.771, with an average of 0.722 per locality (Appendix A). The average observed heterozygosity (Ho) at loci in each locality ranged from 0.328 (KNDG) to 0.384 (BGRB), with an average of 0.356 per locality, while the average expected heterozygosity (He) ranged from 0.370 (SSLI) to 0.431 (CMOE), with an average of 0.402 per locality (Appendix A). Fixation indices (F) at the locality level varied between 0.062 (BK-SG) and 0.269 (CMOE), with an average of 0.166 per locality, and percentage of polymorphic loci (PPL) varied between 80% (HOUET and KNDG) and 100% (CMOE) at localities with an average of 88.6% (Table 3).

### 3.2. Population Structure

#### 3.2.1. Genetic Differentiation

Genetic differentiation was estimated based on an analysis of molecular variance (AMOVA), as well as the coefficient of genetic differentiation (Fst), gene flow (Nm), and Nei genetic distance between populations.

The analysis of molecular variance (AMOVA) showed that 2% of the total variation was observed between populations, 57% was observed between individuals, and 41% was observed within individuals (Figure 2). Analysis of genetic differentiation (Fst) and gene flow (Nm) between population pairs showed low genetic differentiation associated with high gene flow between *B. dorsalis* populations in Burkina Faso. The low Fst (0.009) and the high Nm (27.528) were observed between Comoé and Poni, while the highest Fst (0.023) and the lowest Nm (10.62) were observed between Kénédougou and Sissili (Table 4).

Analysis of Nei’s genetic distance showed little overall genetic distancing between *B. dorsalis* populations at the different study sites (Figure 3). The lowest genetic distance (0.009), represented by the darkest pink color, was observed between Comoé (CMOE) and Poni (PONI), while the largest distance (0.04), represented by the red color, was observed between Kénédougou (KNDG) and Sissili (SSLI). Genetic distancing dendrograms associated with the heatmap grouped the seven *B. dorsalis* sub-populations into clusters (Figure 4). The horizontal dendrogram shows that the Boulkiemdé–Sanguié (BK-SG) and Sissili (SSLI) sub-populations form a cluster distant from that formed by the Comoé (CMOE), Kénédougou (KNDG), Bougouriba (BGRB) and Poni (PONI) sub-populations organized into sub-clusters, grouping Comoé and Kénédougou on the one hand, and Bougouriba and Poni on the other. The vertical dendrogram shows that the Comoé, Kénédougou, and Houet sub-populations together form a cluster with a sub-cluster formed by Comoé and Kénédougou. The Bougouriba, Poni, and Sissili sub-populations together form another cluster that also includes a sub-cluster made up of Sissili and Poni (Figure 3).

#### 3.2.2. Multivariate Analysis and Genetic Clustering

Bayesian admixture analysis of the 10 microsatellite loci using STRUCTURE software showed that the maximum number of clusters (K) corresponding to the highest value of ΔK was estimated at K = 3 (Appendix A). The *B. dorsalis* populations of seven localities in Burkina Faso are structured into three ancestral genetic groups, each with different proportions of ancestral genotype. In our analyses, an individual was considered to belong to a given ancestral group if its ancestral proportion corresponding to this group was the highest. In the population as a whole, 30.8% of individuals belonged to the first ancestral group (red), 49.5% belonged to the second ancestral group (green) and 19.7% belonged to the third ancestral group (blue) (Figure 4A). Within localities, the majority of individuals in the sub-populations of Comoé (CMOE) (51.11%) and Houet (HOUET) (80%) belonged to the second ancestral group, while the minority (22.22% and 6.66%, respectively), belonged to the first ancestral group. The majority of individuals in Bougouriba (BGRB) (42.22%), Boulkiemdé–Sanguié (BK-SG) (55.55%), and Kénédougou (KNDG) (42.22%) belonged to the second ancestral group, while the minority (17.77%, 15.55%, and 26.66%, respectively), belonged to the third ancestral group. The majority of individuals from Poni (PONI) (40%) and Sissili (SSLI) (46.66%) belonged to the first ancestral group, while the minority (26.66% and 11.11%, respectively) belonged to the third ancestral group (Figure 4B).

DAPC analysis of microsatellite data was used to infer the sub-structures of the *B. dorsalis* population in Burkina Faso at K = 3. Cross-validation and the associated numerical methods were used to retain the first 25 PCs (Appendix A) provided by the principal component analysis (PCA) with a conserved variance of 99.7%. The proportions of genetic variance explained by each DAPC axis ranged from 29.8 (axis 1) to 4.1 (axis 6) (Appendix A), and the number of discriminant functions retained was three. The scatterplot plot was generated based on the first two axes, providing a total capture of 54% of the genetic variance observed in the data, with a participation of 29.8% and 24.2% for axis 1 and 2, respectively (Figure 5). This DAPC plot showed a grouping of individuals into three weakly differentiated clusters with Bougouriba (BGRB), Comoé (CMOE), Houet (HOUET), Poni (PONI), and Sissili (SSLI) in a first cluster, Kénédougou (KNDG) in a second cluster, and Boulkiemdé–Sanguié (BK-SG) in a third cluster (Appendix A).

## 4. Discussion

*Bactrocera dorsalis* is currently a serious threat to horticulture in sub-Saharan Africa. Clearly, effective control of this insect pest would require a good understanding of its ecology, biology, population genetics, and other relevant parameters. In sub-Saharan Africa, despite ongoing efforts, much remains to be done in the investigation of this insect pest. Several studies were carried out to inventory the main and intermediate host plants of *B. dorsalis* among cultivated and wild plants, both commercial and non-commercial, to elucidate the seasonal abundance and population dynamics of this insect pest [15,41]. However, data are lacking on a number of parameters, including the distribution of insecticide resistance and genetic diversity and structuring. An initial study by Khamis et al. (2009) looked at the diversity and genetic structuring of *B. dorsalis* populations in West Africa, Central Africa, East Africa, and Sri Lanka [32]. Another study by Qin et al. (2018) gave a global overview of the different structures of *B. dorsalis* across locations in Africa, Asia, and Hawaii. In Africa, 11 countries, including Burkina Faso, were involved [5]. Two other studies by Faye et al. (2020) and Diallo et al. (2021) pointed out the genetic structuring of *B. dorsalis* in certain regions of Senegal [42,43]. This study is the first to provide a specific overview of the genetic structuring of *B. dorsalis* in Burkina Faso.

The results of transversal collections of *B. dorsalis* scales during July 2021 across three types of plant formations observed at seven sampling sites in Burkina Faso showed that the highest proportion of flies was observed in Kénédougou, while the lowest proportion was observed in Sissili. A study in Benin showed that the abundance of *B. dorsalis* was strongly correlated with relative humidity and rainfall [44]. Therefore, the uneven distribution of rainfall would be a factor influencing the distribution of *B. dorsalis* in Burkina Faso. The results showed that the most watered areas, such as Comoé, Bougouriba, Kénédougou, Houet, and Poni in the Sudanian climate, had higher proportions of flies, while less irrigated areas such as Boulkiemdé–Sanguié and Sissili in the Sudano–Sahelian climate, had lower proportions (Table 1). Of the three types of plant formations surveyed, the results showed that a high proportion of flies was observed in mango orchards, while a low proportion was observed in natural formations. In Burkina Faso, the natural formations are composed of plants species such as *Annona senegalensis*, *Landolphia heudelotii*, *Opilia celtidifolia*, *Sarcocephalus latifolius*, *Uvaria chamae*, etc., which are secondary hosts for *B. dorsalis*, whereas mangoes (*Mangifera indica*) are its preferred host fruits [44]. Furthermore, this knowledge of differential distribution of *B. dorsalis* abundance in Burkina Faso would make it possible to better guide the implementation of the various control interventions for greater impact at a national scale.

Analysis of the genetic diversity of *Bactrocera dorsalis* in Burkina Faso through polymorphisms of the ten microsatellite loci showed a high genetic diversity in all sub-populations with an average Shannon Information Index (I) of 0.722. The highest genetic diversity was observed in Comoé (I = 0.771), while the lowest was observed in Sissili (I = 0.657). The same finding was reported by Khamis et al. (2009), who observed similarly high genetic diversity in West Africa (0.59), and this diversity was also slightly higher than that observed in East Africa (0.54) [32]. Qin et al. (2018) also observed high genetic diversity (0.55) in *B. dorsalis* populations in certain African localities, including Burkina Faso [5]. These results show that *B. dorsalis* has successfully adapted in West Africa, and populations are expanding more than fifteen years since its invasion in around 2005 [1,16,20,32,45]. Indeed, the West African region is a major fruit production and export area, more specifically for mango, which is a key host for *B. dorsalis*. In addition, *B. dorsalis* is also known to have a very high capacity for invasion and adaptation [11,32]. Fixation indices (F) in the different sub-populations ranged from 0.062 (Boulkiemdé–Sanguié) to 0.269 (Comoé) with an average of 0.166, showing that all seven *B. dorsalis* sub-populations deviated from the Hardy–Weinberg equilibrium [46] with heterozygote deficiencies. *B. dorsalis* is a highly fertile sexual species in which males and females mate with several partners. Females reproduce several times during their lifetime, lasting around three months. The flies reach sexual maturity in around 30 days after undergoing successive embryonic, larval, pupal, and imaginal development, giving rise to overlapping generations of flies. The species is also known for its great dispersal capacity, often facilitated by commercial trade in fruit [19,47]. Populations of *B. dorsalis* therefore deviate from the Hardy–Weinberg model because of the overlapping generations and migrations that characterize them. The recent introduction of *B. dorsalis* in Burkina Faso could also be a factor contributing to the deviation from the Hardy–Weinberg equilibrium. In fact, newly established populations could generally be limited in size or subject to selection pressure as they adapt to the new environment. The indices of genetic differentiation (Fst) varying from 0.009 (Comoé–Poni) to 0.023 (Kénédougou–Sissili) observed between the sub-populations were low according to Wright’s scale, which considers that an Fst between 0 and 0.05 indicates low genetic differentiation [48]. These low genetic differentiations led to high gene flows, varying from 10.62 between Kénédougou and Sissili to 27.53 between Comoé and Poni. In fact, gene flow is a factor in the adaptation and persistence of the species in new environments through the dispersal of selected resilience traits such as heat- or cold-resistance alleles and insecticide-resistance alleles [24,49]. In addition, local gene flow is a crucial advantage for the success of a genetic control intervention, facilitating the dispersal of the gene of interest [24]. However, it should also be noted that the estimated gene flow could constitute an additional complication in the choice of trial sites for a potential evaluation of a genetically modified strain in the natural environment. Indeed, trial sites with a good level of containment, exchanging fewer or no individuals with the surrounding sites, are ideal candidates. These sites offer settings that minimize the risk of invasion of genetically modified insects into non-target sites and maximize the chances of success by preventing bias in the experiment through potential invasions of surrounding wild strains into the trial sites [50]. The genetic differentiations observed were correlated with the Nei’s genetic distance observed between the sub-populations, ranging from 0.04 between Kénédougou and Sissili to 0.009 between Comoé and Poni. The high flight capacity and variety of host fruits of *B. dorsalis* would facilitate its dispersal from one site to another [32]. These migrations help to reduce the genetic difference and/or the genetic distance between the different populations.

Bayesian admixture analysis in STRUCTURE showed that the *B. dorsalis* populations in Burkina Faso were organized into three genetic groups derived from three ancestral origins. A proportion of 30.8% of the total population belonged to the first ancestral group, 49.5% to the second group, and 19.7% to the third group. This is consistent with the three different invasion events of *B. dorsalis* in Burkina Faso, as suggested by Khamis et al. (2009) for the invasion of Africa [32]. Within the sites, the second ancestral group was more represented in Bougouriba, Boulkiemdé–Sanguié, Comoé, Kénédougou, and Houet, the first ancestral group was more represented in Poni and Sissili, and the third ancestral group was not dominant at any of the sites but had a significant proportion in Comoé, Kénédougou, and Poni. These results show that the population of *B. dorsalis* fruit flies is not geographically structured in Burkina Faso, but it would appear that the second ancestral group is much more dominant in the western part of the country, with Houet as the epicenter, while the first group is more dominant in the southern part.

DAPC is a multivariate analysis used to infer a model of genetic variation providing information on genetic differentiation, both between subgroups and between individuals, while minimizing information within subgroups [37]. The DAPC of *B. dorsalis* in Burkina Faso, using microsatellite analysis, showed that the populations were structured into three main genetic groups that were not very well separated. The first group consisted of individuals from the localities Bougouriba, Comoé, Houet, Poni, and Sissili, the second group consisted of individuals from Kénédougou, and the third group consisted of individuals from Boulkiemdé–Sanguié. Kénédougou, which accounts for almost half of the country’s mango production, is a major production area, exporting fruit to other regions of Burkina Faso and abroad. The area imports relatively few fruits from other regions. Interregional trade in fruit is the main means by which fruit flies, including *B. dorsalis*, are exchanged [51], so the site of Kénédougou constitutes a sort of enclave that could evolve into a special structure. Boulkiemdé–Sanguié is also a site that could evolve into a particular structure due to the geographical distance separating it from the other sites, unlike Kénédougou. Furthermore, the low level of separation between the genetic groups observed demonstrates an absence of geographical structure. This can be attributed to the recent introduction of *B. dorsalis* in Burkina Faso. The genetic structure is also a variable that is strongly influenced by the gene flow factor [24,49].

## 5. Conclusions

The oriental fruit fly, *B. dorsalis*, is a serious threat to food security worldwide. Control strategies for this insect pest in Burkina Faso are mainly based on chemical pesticides, entomopathogenic fungi, parasitoids, predatory ants and, on the horizon, genetic control. Ensuring the effectiveness of these strategies will require a good understanding of this pest, including its biology, distribution, diversity, and genetic structure. This study provides valuable information about the geographical distribution, genetic diversity, and structure of *B. dorsalis* in Burkina Faso. Briefly, the results showed that *B. dorsalis* was unevenly distributed in Burkina Faso due to the unequal distribution of rainfall. The genetic variability was high within populations, the genetic difference was low between them, and there was no evidence of genetic structure. The gene flows between the different populations were high. The information provided on the differential distribution of *B. dorsalis* in Burkina Faso will help to better direct and rationalize control means. The high genetic diversity and the absence of an evident geographical structure could confirm the hypothesis of a foreign invasion of *B. dorsalis* in Burkina Faso. The high level of gene flow between populations is a factor that would favor the dispersal of potential insecticide resistance, and this information could be an asset for managing such concern. In addition, these observed gene flows could be linked to the existing fruit exchange flows between the study sites, which will need to be clarified. The results provided will help guide scientists in the implementation of novel control strategies, including genetic control.

## Figures and Tables

**Figure 1 insects-15-00298-f001:**
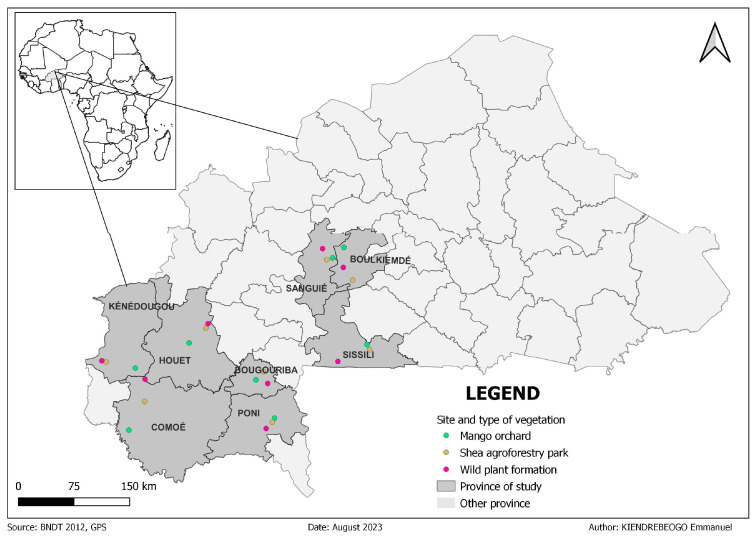
Map showing the sampling sites of *Bactrocera dorsalis*. The green markers represent mango orchards, the yellow markers represent shea agroforestry parks, and the pink markers represent wild plant formations.

**Figure 2 insects-15-00298-f002:**
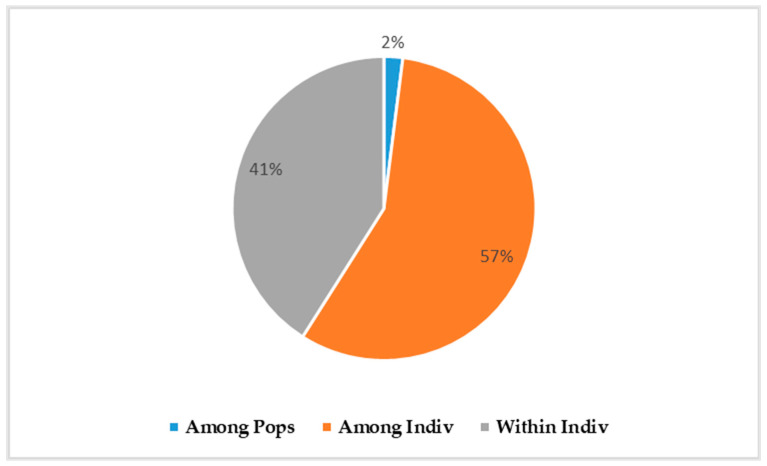
Analysis of molecular variance. The pie chart shows the percentages of molecular variance among populations (Among Pops), among individuals (Among Indiv), and within individuals (Within Indiv). The blue color within the pie chart represents the genetic variation observed between populations (2%), the orange color represents the genetic variation observed between individuals (57%), and the gray color represents the genetic variation observed within individuals (41%).

**Figure 3 insects-15-00298-f003:**
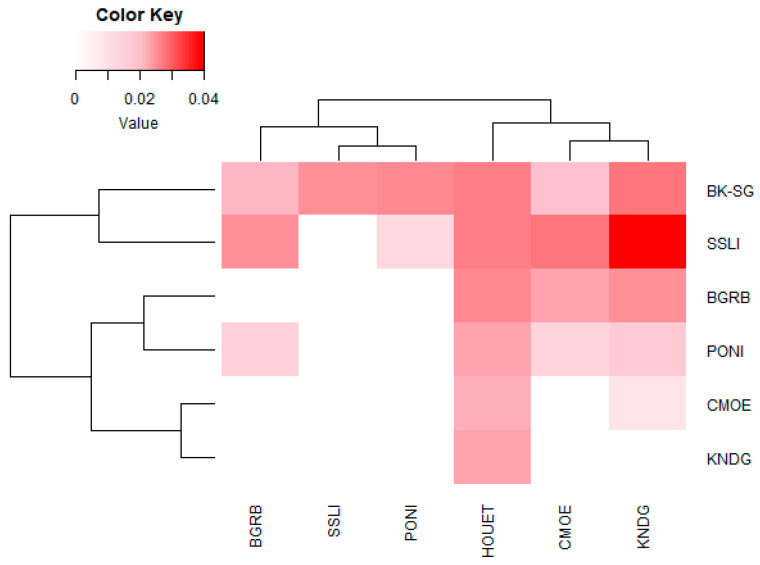
Heatmap of Nei’s Genetic distance (NGD). The intensity of the colors observed between the populations on the heatmap, varying from pink to red, is proportional to the genetic distance separating them. The lowest genetic distance was observed between Comoé (CMOE) and Kénédougou (KNDG), while the highest was observed between Kénédougou (KNDG) and Sissili (SSLI).

**Figure 4 insects-15-00298-f004:**
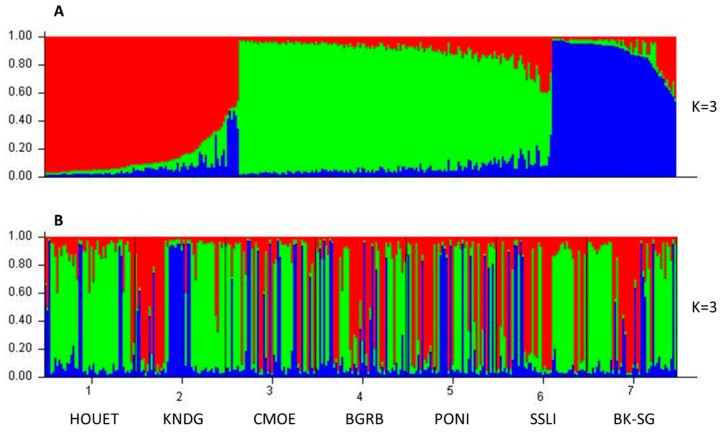
Genetic structuring of *Bactrocera dorsalis* populations based on STRUCTURE analysis at K = 3. (**A**) Sorted by Q plot, (**B**) grouped by pop Id plot. Each vertical bar represents an individual bearing the genotypic proportions of belonging to the different ancestral groups. The first ancestral group (red), the second ancestral group (green) and the third ancestral group (blue). Bougouriba (BGRB), Boulkiemdé–Sanguié (BK-SG), Comoé (CMOE), Houet (HOUET), Kénédougou (KNDG), Poni (PONI), Sissili (SSLI).

**Figure 5 insects-15-00298-f005:**
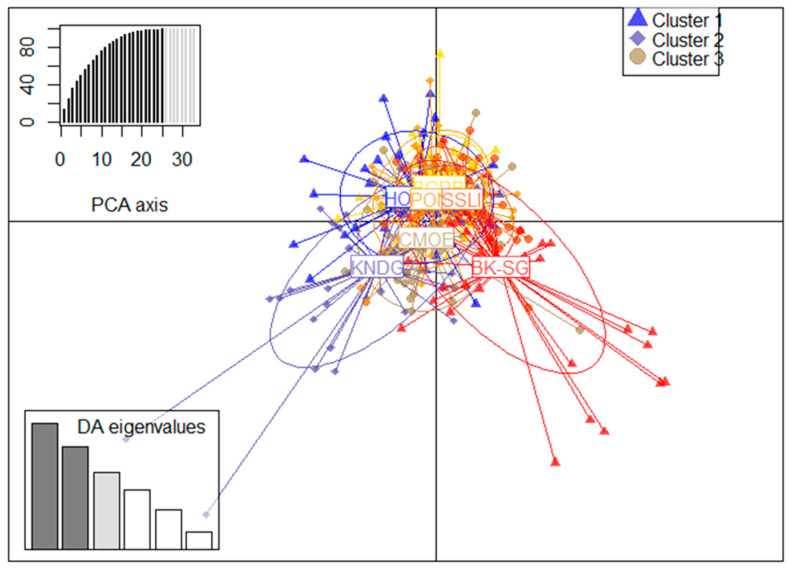
DAPC of *Bactrocera dorsalis* population in Burkina Faso. Bougouriba (BGRB), Boulkiemdé–Sanguié (BK-SG), Comoé (CMOE), Houet (HOUET), Kénédougou (KNDG), Poni (PONI), and Sissili (SSLI). Three weakly differentiated clusters stand out, with individuals from Comoé, Houet, Bougouriba, Poni, and Sissili in the first cluster, individuals from Kénédougou in the second, and individuals from Boulkiemdé in the third. A specific color is assigned to each site and their corresponding PCs as follows: BGRB (yellow), BK-SG (red), CMOE (brown), HOUET (blue), KNDG (purple), PONI (orange), and SSLI (pink). The dark portion of the PCA axis legend diagram represents the first 25 PCs provided by the PCA analysis. The axes of the DAPC that best explain genetic variances are depicted in dark bars in the DAPC eigenvalues legend diagram. The shapes of the markers corresponding to different PCs (triangle, diamond, circle) depend on the corresponding cluster membership.

**Table 1 insects-15-00298-t001:** Distribution of *Bactrocera dorsalis* abundance according to sampling sites and plant formation in Burkina Faso in July 2021.

Sampling Sites	Plant Formations	Density (FTD)/Plant Formation	Proportions (%) [95%Cis]/Plant Formation/Site	Density (FTD)/Site	Proportions (%)/Site
BGRB	MO	38	24.36 [17.62, 31.00]	156	2.97 [02.51, 03.43]
	SAP	50	32.05 [24.73, 39.37]		
	WLD	68	43.59 [35.76, 51.42]		
BK-SG	MO	17	19.10 [10.95, 27.25]	89	1.69 [01.35, 02.04]
	SAP	09	10.11 [03.86, 16.36]		
	WLD	63	70.79 [61.33, 80.25]		
CMOE	MO	504	79.18 [76.03, 82.31]	637	12.14 [11.25, 13.02]
	SAP	92	14.45 [11.71, 17.17]		
	WLD	41	06.43 [04.53, 08.34]		
HOUET	MO	149	83.71 [78.27, 89.15]	178	3.39 [02.99, 03.78]
	SAP	24	13.48 [08.48, 18.48]		
	WLD	05	02.82 [00.39, 05.23]		
KNDG	MO	1124	37.64 [35.99, 39.28]	2986	56.88 [55.55, 58.21]
	SAP	1313	43.97 [42.19, 45.75]		
	WLD	549	18.39 [16.99, 19.76]		
PONI	MO	645	57.69 [54.87, 60.63]	1118	21.29 [20.18, 22.40]
	SAP	92	8.23 [06.62, 09.85]		
	WLD	381	34.08 [31.32, 36.86]		
SSLI	MO	45	51.72 [41.20, 62.24]	87	1.66 [01.31, 01.90]
	SAP	31	35.63 [25.61, 45.64]		
	WLD	11	12.64 [05.66, 19.61]		

Bougouriba (BGRB), Boulkiemdé–Sanguié (BK-SG), Comoé (CMOE), Houet (HOUET), Kénédougou (KNDG), Poni (PONI), Sissili (SSLI), Mango orchard (MO), Sea Agroforestry Park (SAP) and Wild (WLD). Number of flies per trap per day (FTD). Confidence intervals (CI) are indicated in brackets.

**Table 2 insects-15-00298-t002:** Summary of genetic statistics and Wright’s F-statistic for each microsatellite locus.

Locus	N	Na	Ne	I	Ho	He	Ht	Fis	Fit	Fst	Nm
Bd15	44.286	1.286	1.007	0.018	0.003	0.006	0.006	0.491	0.499	0.015	16.819
Bd19	44.286	4	3.131	1.239	0.622	0.673	0.7	0.075	0.111	0.039	6.139
Bi1	44.143	3	1.527	0.611	0.356	0.34	0.345	−0.044	−0.03	0.014	17.553
Bi5	44.429	3.571	1.454	0.594	0.276	0.304	0.31	0.092	0.109	0.019	12.915
Bi8	44.714	2	1.147	0.223	0.035	0.118	0.125	0.704	0.719	0.052	4.561
Bi10	44.143	3	1.834	0.773	0.292	0.447	0.462	0.347	0.368	0.032	7.577
MS12A	43.857	3	1.894	0.787	0.589	0.47	0.477	−0.253	−0.233	0.016	15.846
Bd85b	45	3.714	2.613	1.055	0.438	0.616	0.644	0.288	0.32	0.044	5.368
MS4	44.286	4.429	3.013	1.235	0.791	0.664	0.68	−0.191	−0.164	0.023	10.732
MS3	44	3.143	1.623	0.688	0.162	0.378	0.384	0.571	0.577	0.015	16.091
Mean	44.314	3.114	1.924	0.722	0.356	0.402	0.413	0.208	0.228	0.027	11.36
SE	0.111	0.130	0.089	0.047	0.031	0.026	0.048	0.103	0.101	0.004	1.623

Effective population size (N), average number of alleles (Na), average effective number of alleles (Ne), Shannon’s Information Index (I), mean observed heterozygosity (Ho), mean expected heterozygosity (He), total expected heterozygosity (Ht), inbreeding coefficient (Fis) and total inbreeding coefficient (Fit), genetic differentiation indices (Fst), gene flow (Nm), standard error (SE).

**Table 3 insects-15-00298-t003:** Genetic diversity parameters among the seven populations of *Bactrocera dorsalis*.

Pop	N	Na	Ne	I	Ho	He	uHe	F	PPL
HOUET	44.6	3.3	1.895	0.718	0.347	0.390	0.394	0.128	80%
KNDG	44.8	3.5	1.963	0.741	0.328	0.397	0.402	0.202	80%
CMOE	43.2	3.2	2.032	0.771	0.348	0.431	0.436	0.269	100%
BGRB	44.6	3.1	1.916	0.740	0.384	0.423	0.428	0.167	90%
PONI	44.5	2.9	1.926	0.716	0.365	0.398	0.402	0.159	90%
SSLI	44.6	2.8	1.819	0.657	0.354	0.370	0.374	0.164	90%
BK-SG	43.9	3.0	1.919	0.712	0.369	0.403	0.407	0.062	90%
Mean	44.3	3.1	1.924	0.722	0.356	0.402	0.406	0.166	88.6%

Effective population size (N), average number of alleles (Na), average effective number of alleles (Ne), Shannon’s Information Index (I), mean observed heterozygosity (Ho), mean expected heterozygosity (He), unbiased expected heterozygosity (uHe), percentage of polymorphic loci (PPL).

**Table 4 insects-15-00298-t004:** Pairwise Nm (above diagonal) and Fst (below diagonal) values among the seven populations of *Bactrocera dorsalis*.

Pop	BGRB	BK-SG	CMOE	HOUET	KNDG	PONI	SSLI
BGRB		16.416	20.583	11.114	12.250	22.477	12.908
BK-SG	0.015		17.607	11.655	11.114	13.639	12.250
CMOE	0.012	0.014		14.456	22.478	27.528	13.639
HOUET	0.022	0.021	0.017		14.456	13.639	14.456
KNDG	0.020	0.022	0.011	0.017		17.607	10.620
PONI	0.011	0.018	0.009	0.018	0.014		24.75
SSLI	0.019	0.020	0.018	0.017	0.023	0.010	

## Data Availability

The original data presented in the study are openly available in [pCloud Drive] at [https://u.pcloud.link/publink/show?code=XZNObm0Z4ljMuPWUHE574LBAt1hSPkEnAT3y] accessed on 17 April 2024.

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
