# Peer review of "Genetic Diversity and Population Structure of the Invasive Oriental Fruit Fly, Bactrocera dorsalis (Diptera: Tephritidae) in Burkina Faso"

_insects, 2024, doi:10.3390/insects15050298_

Round 1
Reviewer 1 Report
Comments and Suggestions for Authors
The paper is very interesting especially because it deals with one of the worst pests in the world and its genetic resulted to be very complex due the biological characteristics B. dorsalis has. Based on the present study, carried out in 7 Burkinabè sites divided in 3 plant formations, authors highlighted, through the use of microsatellites, the genetic analysis of Bactrocera dorsalis. The study revealed high genetic diversity across all subpopulations. Fixation indices indicated deviations from the Hardy-Weinberg equilibrium due to overlapping generations and migrations. Low genetic differentiations were observed between subpopulations, leading to high gene flows. Bayesian admixture analysis showed three genetic groups derived from ancestral origins and Discriminant analysis identified three principal clusters with moderate separation. Furthermore, the study highlighted that two sites (KNDG and BK-SG ) showed potential for unique genetic structures (special structure) due to their geographical locations and limited import of fruits.
The manuscript is well-written and comprehensive. Below, I have provided some suggestions and clarifications for the authors.
Introduction
Line 50: what does mean “s.l.”? I know it means “sensu latu”, but I think authors should explain the meaning in the first mention. Furthermore, authors should give some reference to the statement: “comprises around a hundred morphologically 50 similar species.” In some cases, species included in the complex are higher or lower.
Line 53: although they are synonymous, is crucial the use of complete name in the first mention of the species.
57- please delete the authority
M&M
Lines 93-98 and 110. Authors use 45 flies per sites, there are 7 sites and in each site are included 3 plant formations (mango orchards (MO), shea agroforestry parks (SAP), and natural formations (WLD)). Does it mean that for each plant formation the author selected 15 flies? This is not clear because in line 110 authors stated “randomly” and this could be misleading.
Furthermore, I am doubtful about the choice of the sample size. Potentially, the authors may not be certain that they have taken all the variables of the population into account. To assess the highest number of variables a possible solution could be found in M&M paragraph of De Benedetta et al. (2022) https://doi.org/10.1038/s41598-022-23520-2. However, if authors applied a different reasoning, in my opinion, it should be reported.
Lines 113-114: why did the author choose these Microsatellites? I think the reason is crucial.
GenAlEx version 6.51b2 was mentioned both in 2.3.1 both in 2.3.2. Could the author merge the paragraphs?
Authors gave reason about the choose of the month of July to collect insect in Discussions paragraph, I think this part should be moved in M&M section
Results
Lines 171-185: author should resume the paragraph in a table. I think it could simplified the readers’ understanding.
203-207: information about Shannon index should be reported in M&M paragraph
I believe that if the authors manage to address or clarify these minor shortcomings, the paper should deserve publication.
Reviewer 2 Report
Comments and Suggestions for Authors
The article by Traoré et al. reporting the results of a genetic and populational analysis of Bactrocera dorsalis population samples in Burkina Faso is comprehensive, clearly written and the obtained data are also clearly described. However I would like to comment some points.
L13, L25: The authors declare that B. dorsalis has a worldwide distribution. Does this species occur on the American Continent? (see L57 -L63);
L17, L29 (and across the text): The authors described the collecting sites as “transversally” in Burkina Faso. Transversally would suggest that there was a cline in the collection sites but the map (Fig. 1) shows that a better description would to say that the collection were made in sites at “western areas of the country”;
L73-L75. Why not IIT (Incompatible Insect Technique) is mentioned among the other methods of populational control?
L85-L90: In this last paragraph it must be clearly stated that the previous studies (20,27) were also made with B. dorsalis, etc., etc.
L99: Are the authors sure that methyl-eugenol is “specific” for B. dorsalis?
L110: change: “including” by “among which”, since the 45 males were included in the 315 males; males were randomly “sampled” not “selected”;it is not possible randomly select.
L134-L137: This paragraph seems not to be necessary since the information is already described in the previous one (L126-132);
L195-L216: In the last line (216) of this section Table 1 is correctly called. However, at the beginning of the section, L197-L199, it is described the total number of alleles, etc. However, these values are not shown in Table 1. Hence the text must be corrected: after the words “….31.14 per locality.” The next phrase could start as “As shown in Table 1, the mean…
L203-207: This sentence describing the Shannon Indexes would be better located in Materials & Methods;
L217 and L 229: In both Tables what is N?
In the subtitles at the bottom of both Tables it must be included the full names of the codes, Average number of alleles (Na), Average effective number of alleles (Ne), etc.
L363: “….years of evolution. Are the authors sure that evolutionary modifications have occurred during this period?
L370-372: This sentence shall be deleted. It is no longer necessary to describe what is the Hardy-Weinberg equilibrium!
L377-379: Could not also be explained the absence of H-W equilibrium by the recent events of introduction of the species?
L389: Substitute the “observed” by “estimated”;
L441: “diversified” is dubious: it may indicate that populations are distinct one from another or that both shows variations although they could be similar. Must be explicitly!
